# Research on Three-Dimensional Stress Monitoring Method of Surrounding Rock Based on FBG Sensing Technology

**DOI:** 10.3390/s22072624

**Published:** 2022-03-29

**Authors:** Minfu Liang, Xinqiu Fang, Yang Song, Shuang Li, Ningning Chen, Fan Zhang

**Affiliations:** 1School of Mines, China University of Mining and Technology, Xuzhou 221116, China; liangmf2014@cumt.edu.cn (M.L.); ts15020009a3tm@cumt.edu.cn (N.C.); ts21020199p21@cumt.edu.cn (F.Z.); 2Research Center of Intelligent Mining, China University of Mining and Technology, Xuzhou 221116, China; 3School of Economics and Management, China University of Mining and Technology, Xuzhou 221116, China; lishuangchina@cumt.edu.cn

**Keywords:** surrounding rock stress, three-dimensional stress sensors, fiber Bragg grating (FBG), finite element analysis (FEA), performance testing

## Abstract

Research on the stress state of rock mass is essential for revealing the distribution characteristics and evolution law of the surrounding rock stress field in the roadway, studying the coal–rock dynamic disaster and the design of roadway support. This thesis proposes a three-dimensional stress monitoring method for surrounding rocks based on fiber Bragg grating (FBG) sensing technology and a cube-shaped three-dimensional stress fiber grating sensor is developed based on the principle of this monitoring method. According to the fiber grating strain obtained by numerical simulation, the calculated three-dimensional stress value is basically consistent with the theoretical value. The margin of error was plus or minus one percentage point. The sensing performance of the sensor was tested using a uniaxial compression experiment instead of a triaxial compression experiment. The experimental results show that in the range of 0~50 Mpa, the sensor’s sensitivity to *X*, *Y* and *Z* axis stress is 25.51, 25.97 and 24.86 pm/Mpa, respectively. The relative error of measured stress is less than 4%. Meanwhile, the sensor has good linearity and repeatability, and has broad application prospects in the field of underground engineering safety monitoring such as in coal mines and tunnels.

## 1. Introduction

With the increase of the coal mining depth in China, the surrounding rock stress also increases, which leads to an increase in the frequency of accidents such as rock bursts, coal and gas outbursts and other accidents [1,2,3,4]. Real-time monitoring of surrounding rock stress, as one of the effective means to prevent coal–rock dynamic disaster accidents, is helpful to grasp the stress distribution and stress variation law of surrounding rock in the process of coal mining and to evaluate the risk of the working face area and optimize the parameters of roadway support. Therefore, the three-dimensional stress monitoring of the surrounding rock is very important.

At present, in coal mine surrounding rock stress monitoring [5,6,7], one-dimensional stress sensors are mostly used to monitor vertical stress or uniaxial stress and three-dimensional stress distribution cannot be obtained. The measurement of the three-dimensional stress of the surrounding rock can use the in-situ stress measurement method, such as the hydraulic fracturing method [8] and the stress relief method [9]. However, during the coal mining process, the magnitude and direction of surrounding rock stress changes. These methods cannot accurately obtain the three-dimensional stress field distribution of the rock surrounding the roadway. Hydraulic fracturing method is a plane stress measurement method, which can only measure the two-dimensional stress in the borehole plane. It can be used to measure the direction and size of the maximum and minimum principal stress, and it is more complex in practical operation. The core element of the stress relief method is the traditional electrical strain gauge, which is greatly affected by external factors such as material elastic modulus, Poisson’s ratio and borehole diameter. It is vulnerable to electromagnetic interference and poor working durability. The correction coefficient must be added to determine the accuracy of the test, which cannot realize long-term continuous monitoring and cannot meet the needs of mine safety monitoring.

Fiber grating sensing technology has the advantages of its small size, anti-electromagnetic interference, high sensitivity, good corrosion resistance, great waterproof performance and easy networking. Therefore, this technology is widely used in aviation, electric power, transportation, construction, coal mining and other fields [10,11,12,13,14,15]. Recently, some scholars have developed some three-dimensional force sensors based on FBG sensing technology but most of them are applied in other fields. For example, Hui-Chao Xu proposed a three-dimensional force sensor based on multiple FBG strain sensors for building structure health monitoring [16]; Yongxing Guo investigated a three-axis FBG force sensor for sensing the force exerted by a robot finger on an object [17]; Sonnenfeld Camille exploited the directional dependence of the lateral sensitivity of the MOFBG to construct a sensing device for measuring strain along the three main mechanical directions of laminated composites [18]. Zhi Zhou developed a fiber-grating-based 3D strain sensor for highway and civil structure health monitoring [19]. In the mining field, Liang Minfu proposed an FBG inclination sensor to monitor the attitude of hydraulic support [20]; Tao Zhigang monitors coal and rock movement in real time based on a quasi-distributed FBG sensor, which provides a basis for pillar-free mining [21]; Wang Peng designed an FBG force measuring bolt to continuously monitor the roof pressure of the coal roadway [22]; Chai Jing uses an FBG sensor to monitor the structural deformation of key layers at different stages and finally realizes the inversion of overburden state in goaf [23]; Wan Xiaorong developed an FBG stress sensor to predict the stability after coal roadway excavation [24]. However, the application of FBG three-dimensional stress sensors in coal mine engineering has not been reported, so it is significance to study the application of three-dimensional stress sensors in the surrounding rocks.

In this paper, a three-dimensional stress sensor based on FBG sensing technology is proposed to monitor the three-dimensional stress in the surrounding rock. The feasibility of the sensor is verified by theoretical analysis, simulation and experiment. This research is of great significance for revealing the distribution characteristics and evolution law of the surrounding rock stress field in the roadway, prevention and control of coal and rock dynamic disasters, optimization of roadway support parameters, risk assessment of working face areas and realization of mine safety production. The sensor proposed in this paper has great engineering application value and application prospects in the field of underground engineering safety monitoring.

## 2. Theoretical Analysis and Structural Design of Sensors

### 2.1. Basic Theory and Sensing Characteristics of FBG

FBG sensing technology takes optical fibers and light waves as the medium and carrier, respectively. Optical fiber is a cylindrical medium for transmitting light waves, consisting of a core, a cladding and a coating layer. Because the refractive index n_1_ of the core is larger than the refractive index n_2_ of the cladding, the light wave can propagate along the core when the total reflection of light occurs. FBG places the optical fiber under an ultraviolet light source with periodic spatial variation to make the refractive index change of the fiber core.

Figure 1 shows the working principle of the FBG sensing technology. The manufacture of fiber Bragg gratings is based on the photosensitivity of the fiber; an FBG is equivalent to a wavelength filter. According to the Maxwell’s equations and coupled-mode theory of optical fibers, the Bragg wavelength of reflection was given by [25]:(1)$$\lambda_{B}=2n_{eff}\Lambda$$
where *λ_B_* is the reflective center wavelength of a fiber Bragg grating, *n_eff_* is the effective refractive index and Λ is the grating period. When the Bragg wavelength satisfies Equation (1), the incident light will be reflected back by the grating. As evident from Equation (1), the Bragg wavelength is shifted if the effective refractive index or the grating period is changed, and both can be directly influenced by strain and ambient temperature. The relationship between the FBG wavelength shift with strain and temperature can be expressed as [26]:(2)$$\frac{\Delta \lambda_{B}}{\lambda_{B}}=\left(1-p_{e}\right)\varepsilon_{z}+\left(\alpha +\xi \right)\Delta T,$$
where Δ*λ* is the variation of the FBG central wavelength, *ε_z_* is the axial strain of the fiber caused by tensile stress, pressure or bending, Δ*T* is the change in ambient temperature, *p_e_*, *α* and *ξ* are effective photo-elastic constants, thermal expansion coefficient and thermo optical coefficient of the fiber, respectively.

When the ambient temperature is constant, the relationship between the FBG wavelength shift and strain can be expressed as [27]:(3)$$\varepsilon =\frac{\Delta \lambda }{\lambda_{B}(1-p_{e})},$$

### 2.2. Surface-Mounted Fiber Grating Strain Transfer Principle

The fiber grating is fixed on the surface of the substrate by sticking. When the matrix is deformed, the fiber grating also deforms, but the strain transfer between them will be partially absorbed by the adhesive layer. The relationship between the axial strain of the FBG and the axial strain of the matrix can be deduced by an elastic mechanics analysis of the fiber micro-element, which is expressed as follows [28]:(4)$$\varepsilon_{f}(x)=\varepsilon_{m}\left[1-\frac{\cosh\left(kx\right)}{\cosh\left(kL\right)}\right],$$
where *x* is the distance from the center point of the length of the fiber adhesive layer, $\varepsilon_{f}$ is the axial strain of the fiber, $\varepsilon_{m}$ is the axial strain of the matrix, 2*L* is the length of the fiber adhesive layer, $k=\sqrt{\frac{DE_{C}}{2\pi E_{f}r_{f}^{2}(r_{m}-r_{f})(1+\mu_{c})}}$. Among them, *D* is the width of the adhesive layer, $E_{C}$ is the elastic modulus of the adhesive layer, $E_{f}$ is the elastic modulus of the optical fiber, $r_{f}$ is the radius of the optical fiber layer, $r_{m}$ is the radius of the matrix layer, and $\mu_{c}$ is the Poisson’s ratio of the adhesive layer.

The average strain transfer efficiency between the FBG and the matrix can be expressed by the ratio of the average strain of the FBG to the matrix measured in the fiber bonding length range, as follows [29,30]:(5)$$\alpha =\frac{\bar{\varepsilon_{f}}}{\bar{\varepsilon_{m}}}=\frac{\frac{2\int_{0}^{L}\varepsilon_{f}(x)dx}{2L}}{\frac{2\int_{0}^{L}\varepsilon_{m}(x)dx}{2L}}=1-\frac{\sinh\left(kL\right)}{kL\cosh\left(kL\right)},$$

From the data in Table 1, the relationship between the ratio of optical fiber strain to matrix strain (i.e., strain transfer efficiency) and the length of the fiber adhesive layer can be calculated, as shown in Figure 2:

It can be seen from Figure 2 that the highest strain transfer efficiency of FBG is in the middle part of the fiber adhesive layer. As it is close to the boundary of the fiber adhesive layer, the strain transfer efficiency decreases gradually and decreases sharply at the end.

### 2.3. Three-Dimensional Stress Sensor Stress Measurement Principle

OA is any straight line in space. As shown in Figure 3, *l*, *m*, and *n* are the cosines of the straight line OA in the *X*, *Y*, and *Z* directions, respectively:(6)$$\left\{ \begin{array}{l} l=\sin\delta \cos\varphi \\ m=\sin\delta \sin\varphi \\ n=\cos\delta \end{array}\right.,$$
where *δ* is the angle between the straight line OA and the *Z* coordinate axis, and *φ* is the angle between the projection of the line OA in XOY plane and the *X* coordinate axis.

The strain state of O at any point in the surrounding rock is $\varepsilon_{j}=\left\{\varepsilon_{x},\varepsilon_{y},\varepsilon_{z},\gamma_{xy},\gamma_{yz,}\gamma_{xz}\right\}$, according to the formula of linear strain in any direction of elastic mechanics space, the linear strain in OA direction is [30]:(7)$$\varepsilon =\varepsilon_{x}l^{2}+\varepsilon_{y}m^{2}+\varepsilon_{z}n^{2}+\gamma_{xy}lm+\gamma_{yz}mn+\gamma_{xz}ln,$$

As can be seen from the above, the line strains in six different directions can characterize the point strain state:(8)$$\varepsilon_{i}=\varepsilon_{x}l_{i}{}^{2}+\varepsilon_{y}m_{i}{}^{2}+\varepsilon_{z}n_{i}{}^{2}+\gamma_{xy}l_{i}m_{i}+\gamma_{yz}m_{i}n_{i}+\gamma_{xz}l_{i}n_{i},$$
where *i* = 1, 2, 3, 4, 5, 6.

The mapping relationship from the strain state component at a point to the line strains in six different directions, passing through this point is as follows:(9)$$\left[\begin{array}{l} \varepsilon_{1}\\ \varepsilon_{2}\\ \varepsilon_{3}\\ \varepsilon_{4}\\ \varepsilon_{5}\\ \varepsilon_{6} \end{array}\right]=\left[\begin{array}{l} l_{1}^{2}\;m_{1}^{2}\;n_{1}^{2}\;l_{1}m_{1}\;m_{1}n_{1}\;l_{1}n_{1}\\ l_{2}^{2}\;m_{2}^{2}\;n_{2}^{2}\;l_{2}m_{2}\;m_{2}n_{2}\;l_{2}n_{2}\\ l_{3}^{2}\;m_{3}^{2}\;n_{3}^{2}\;l_{3}m_{3}\;m_{3}n_{3}\;l_{3}n_{3}\\ l_{4}^{2}\;m_{4}^{2}\;n_{4}^{2}\;l_{4}m_{4}\;m_{4}n_{4}\;l_{4}n_{4}\\ l_{5}^{2}\;m_{5}^{2}\;n_{5}^{2}\;l_{5}m_{5}\;m_{5}n_{5}\;l_{5}n_{5}\\ l_{6}^{2}\;m_{6}^{2}\;n_{6}^{2}\;l_{6}m_{6}\;m_{6}n_{6}\;l_{6}n_{6} \end{array}\right]\left[\begin{array}{l} \varepsilon_{x}\\ \varepsilon_{y}\\ \varepsilon_{z}\\ \gamma_{xy}\\ \gamma_{yz}\\ \gamma_{xz} \end{array}\right],$$

The above is abbreviated as follows:(10)$$\left\{\varepsilon_{i}\right\}=T\times \left\{\varepsilon_{j}\right\},$$
where *j* = *x*, *y*, *z*, *xy*, *yz*, *xz*, ***T*** is the coefficient matrix.

In summary, the relationship between the strains *ε*_1_, *ε*_2_, *ε*_3_, *ε*_4_, *ε*_5_, *ε*_6_ and *ε_x_*, *ε_y_*, *ε_z_*, *γ_xy_*, *γ_yz_*, *γ_xz_* measured by the FBG three-dimensional stress sensing structure is as follows:(11)$$\left\{ \begin{array}{l} \varepsilon_{x}=\varepsilon_{1}\\ \varepsilon_{y}=\varepsilon_{2}\\ \varepsilon_{z}=\varepsilon_{3}\\ \gamma_{xy}=\varepsilon_{1}+\varepsilon_{2}-2\varepsilon_{4},\\ \gamma_{yz}=\varepsilon_{2}+\varepsilon_{3}-2\varepsilon_{5}\\ \gamma_{xz}=\varepsilon_{1}+\varepsilon_{3}-2\varepsilon_{6} \end{array}\right.$$

According to the physical equation of elasticity, the three-dimensional stress is expressed as follows:(12)$$\left\{ \begin{array}{l} \sigma_{x}=\frac{E}{1+\mu }(\frac{\mu }{1-2\mu }\theta +\varepsilon_{x})=\frac{E}{1+\mu }(\frac{1-\mu }{1-2\mu }\varepsilon_{1}+\frac{\mu }{1-2\mu }(\varepsilon_{2}+\varepsilon_{3}))\\ \sigma_{y}=\frac{E}{1+\mu }(\frac{\mu }{1-2\mu }\theta +\varepsilon_{y})=\frac{E}{1+\mu }(\frac{1-\mu }{1-2\mu }\varepsilon_{2}+\frac{\mu }{1-2\mu }(\varepsilon_{1}+\varepsilon_{3}))\\ \sigma_{z}=\frac{E}{1+\mu }(\frac{\mu }{1-2\mu }\theta +\varepsilon_{z})=\frac{E}{1+\mu }(\frac{1-\mu }{1-2\mu }\varepsilon_{3}+\frac{\mu }{1-2\mu }(\varepsilon_{1}+\varepsilon_{2}))\\ \tau_{xy}=\frac{E}{2(1+\mu )}\gamma_{xy}=\frac{E}{2(1+\mu )}(\varepsilon_{1}+\varepsilon_{2}-2\varepsilon_{4})\\ \tau_{yz}=\frac{E}{2(1+\mu )}\gamma_{yz}=\frac{E}{2(1+\mu )}(\varepsilon_{2}+\varepsilon_{3}-2\varepsilon_{5})\\ \tau_{zx}=\frac{E}{2(1+\mu )}\gamma_{xz}=\frac{E}{2(1+\mu )}(\varepsilon_{1}+\varepsilon_{3}-2\varepsilon_{6}) \end{array}\right.,$$
where *E* is the elastic modulus of the sensor matrix, *μ* is the Poisson’s ratio of the sensor substrate material and $\theta =\varepsilon_{x}+\varepsilon_{y}+\varepsilon_{z}$ is the volumetric strain.

Under the condition of constant temperature, the relationship between the reflected wavelength change of the fiber grating and the strain of the fiber grating can be expressed by Equation (3). By combining this with Equations (5) and (12), the relationship between the reflected wavelength variation of the sensing structure fiber grating and the three-dimensional stress at the point can be obtained as:(13)$$\left[\begin{array}{l} \Delta \lambda_{1}\\ \Delta \lambda_{2}\\ \Delta \lambda_{3}\\ \Delta \lambda_{4}\\ \Delta \lambda_{5}\\ \Delta \lambda_{6} \end{array}\right]=\frac{1-p_{e}}{2E}\left[\begin{array}{cccccc} 2\lambda_{B_{1}}\alpha_{1} & -2\mu \lambda_{B_{1}}\alpha_{1} & -2\mu \lambda_{B_{1}}\alpha_{1} & 0 & 0 & 0\\ -2\mu \lambda_{B_{2}}\alpha_{2} & 2\lambda_{B_{2}}\alpha_{2} & -2\mu \lambda_{B_{2}}\alpha_{2} & 0 & 0 & 0\\ -2\mu \lambda_{B_{3}}\alpha_{3} & -2\mu \lambda_{B_{3}}\alpha_{3} & 2\lambda_{B_{3}}\alpha_{3} & 0 & 0 & 0\\ (1-\mu )\lambda_{B_{4}}\alpha_{4} & (1-\mu )\lambda_{B_{4}}\alpha_{4} & -2\mu \lambda_{B_{4}}\alpha_{4} & -2(1+\mu )\lambda_{B_{4}}\alpha_{4} & 0 & 0\\ -2\mu \lambda_{B_{5}}\alpha_{5} & (1-\mu )\lambda_{B_{5}}\alpha_{5} & (1-\mu )\lambda_{B_{5}}\alpha_{5} & 0 & -2(1+\mu )\lambda_{B_{5}}\alpha_{5} & 0\\ (1-\mu )\lambda_{B_{6}}\alpha_{6} & -2\mu \lambda_{B_{6}}\alpha_{6} & (1-\mu )\lambda_{B_{6}}\alpha_{6} & 0 & 0 & -2(1+\mu )\lambda_{B_{6}}\alpha_{6} \end{array}\right]\left[\begin{array}{l} \sigma_{x}\\ \sigma_{y}\\ \sigma_{z}\\ \tau_{xy}\\ \tau_{yz}\\ \tau_{zx} \end{array}\right]$$
where *α_n_* (*n* = 1, 2, 3, 4, 5, 6) is the individual fiber grating strain transfer efficiency. The stress sensitivity of the sensor in the *X*, *Y*, and *Z* directions is $\frac{\lambda_{B_{1}}\alpha_{1}(1-p_{e})}{E}$, $\frac{\lambda_{B_{2}}\alpha_{2}(1-p_{e})}{E}$, $\frac{\lambda_{B_{3}}\alpha_{3}(1-p_{e})}{E}$. Calculated according to the parameters given in Table 1, the sensitivities are 26.5 pm/Mpa, 26.6 pm/Mpa, 26.7 pm/Mpa. The three-dimensional stress sensor designed in this paper can obtain the three-dimensional stress parameters by measuring the wavelength changes of six fiber gratings and substituting them into Equation (13).

### 2.4. FBG Three-Dimensional Stress Sensor Structure Design

It can be seen from the analysis in Figure 4 that the three-dimensional stress state at one point of the surrounding rock includes three normal stresses and three shear stresses; a total of six stress components, namely *σ_x_*, *σ_y_*, *σ_z_*, *τ_xy_*, *τ_xz_*, *τ_yz_*. Considering that at least six linear strains in different directions need to be measured to obtain a one-point three-dimensional stress state, the shape of the sensor matrix is set as a cube; the fiber gratings are arranged along the sensor, diagonal in each coordinate plane of the sensor cube, to measure the strain. The fiber gratings are arranged along the three-dimensional coordinate axes, *X*, *Y*, and *Z* directions, to measure the strain in the center of the three surfaces parallel to the coordinate axis, so that the strain measurement in six different directions can be realized. A structure diagram of the FBG 3D stress sensor is shown in Figure 4.

## 3. Numerical Analysis and Finite Element Simulation

### 3.1. Analysis of Factors Influencing Sensor Sensitivity

The commonly used fiber grating center wavelength and effective elastic-optic coefficient can be regarded as constants. During the analysis, we took *λ_B_* = 1550 nm and *p_e_* = 0.216. The measurement sensitivity of the sensor is mainly affected by the elastic modulus of the sensor matrix and the strain transfer efficiency of the fiber grating. At the same time, the strain transfer efficiency is affected by the fiber bonding length, width, elastic modulus of the binder, Poisson’s ratio and so on.

As can be seen from Figure 5a, the sensitivity of the sensor is inversely proportional to the elastic modulus of the matrix material. When the elastic modulus of the matrix material is less than 40 GPa, the sensitivity of the sensor decreases rapidly with the increase of the elastic modulus of the matrix. However, when it is greater than 40 GPa, the sensitivity of the sensor changes slowly with the increase of the elastic modulus of the matrix. It can be seen from Figure 5b that the influence of the properties of the adhesive layer on the sensitivity of the sensor is smaller than that of the matrix elastic modulus. The sensor sensitivity increases with the increase in the elastic modulus of the adhesive layer, but the growth rate decreases gradually, and decreases as the Poisson’s ratio of the adhesive layer increases. It can be seen from Figure 5c that the sensor sensitivity increases as the width of the adhesive layer increases. When the width of the adhesive layer reaches 3 mm, the sensor sensitivity slows down with the increase of the width of the adhesive layer and then gradually becomes stable; the sensor sensitivity grows with the increase of the optical fiber pasting length. When the pasting length reaches 40 mm, the sensor sensitivity increases slowly as the optical fiber pasting length increases.

### 3.2. Finite Element Analysis of Sensor Sensing Characteristics

In order to verify the sensing characteristics of the sensor, ANSYS Workbench 15.0 is used to simulate the strain distribution in each part of the three-dimensional stress sensor under different pressures; substituting the numerical value obtained from the simulation into the principle formula of the three-dimensional stress sensor to obtain the pressure value. The validity and rationality of its structure can be verified through the above data.

The sensor structure base is a cube with a side length of 40 mm. Semicircular grooves with a diameter of 1 mm are opened on the six surfaces of the substrate, respectively, and six optical fibers are encapsulated in the grooves. The spatial relationship is shown in Table 2. The fiber diameter is 0.245 mm. The difference between the coating and the fiber material is ignored in the experimental simulation. The substrate material is magnesium–aluminum alloy. The adhesive layer uses epoxy resin. Sensor material parameters are shown in Table 3. Finally, the finite element analysis model of the sensor is established in turn, as shown in Figure 6. The total number of model elements is 141,495 and the number of nodes is 324,594. A pressure load of 0~50 MPa is applied to the front of the sensor model in the *X*-axis direction, with a variation of 5 MPa per load stage. The displacement constraint is applied on the negative side of the *X*-axis of the model, that is, the model can deform in the *Y*-axis and *Z*-axis directions, which is similar to the boundary conditions of the uniaxial compression experiment.

Figure 7 shows the equivalent strain of the sensor under a 50 MPa uniaxial load. It can be seen from the figure that the grooving on the surface of the sensor matrix has little effect on the strain distribution of the sensor. The factor that has a great influence on the strain distribution of the sensor is the increase of the strain of the matrix under the groove; the strain of the epoxy resin inside the groove is slightly larger than the strain of the optical fiber and the matrix.

The axial strain of each fiber is derived by setting the Path in the model. The strain value at the center of each optical fiber is the strain value measured by the grating at that location. The variation of the FBG strain with the *X*-axis load is shown in Figure 8.

As shown in Figure 8, the fiber grating on the corresponding axis generates the strain when the sensor is loaded in a single axis. At the same time, the five fiber gratings in other directions are also deformed due to the deformation of the matrix, and the strain value of each fiber grating has a linear relationship with the uniaxial load.

Hydrostatic pressure ranging from 0 MPa to 50 MPa is applied to the sensor model, with each load leave being 5 MPa. Figure 9 shows the equivalent strain of the sensor under a hydrostatic pressure of 50 MPa. It can be seen from the figure that the stress distribution of the sensor substrate is uniform and the grooving on the surface of the sensor matrix has little effect on the strain distribution of the sensor. Meanwhile, the strain of the epoxy resin inside the groove is slightly smaller than the strain of the optical fiber and the matrix.

Figure 10 shows the axial strain distribution of each optical fiber under 50 MPa of hydrostatic pressure. The axial strain distribution of each fiber is basically consistent with the theoretical analysis. The axial strain values of each fiber show a decreasing trend from the middle to both ends and the strain decreases sharply at the end of the fiber. This result is consistent with the theoretical derivation of the strain transfer efficiency of the fiber gratings.

By inserting the Path into the model, the measured strain value at the center of each fiber is the fiber grating strain. Under 50 MPa of hydrostatic pressure, the strains of fiber gratings are 334.79, 334.26, 334.73, 333.82, 334.08 and 333.87 *με*, respectively. From Equation (13), the triaxial pressures of model *X*, *Y* and *Z* can be calculated to be 49.9918 MPa, 49.9908 MPa and 49.9898 MPa, respectively. It can be seen that the calculated value of this model is slightly smaller than the theoretical value but the error is less than 1%, which meets the accuracy requirements of the three-dimensional stress monitoring of the surrounding rock.

## 4. Sensor Experiments and Analysis of Results

### 4.1. Sensor Experiment System

The calibration and testing system of the three-dimensional stress sensor is shown in Figure 11. In the experiments, the strain gauges were pasted on the sensor surface along the axial direction of the fiber grating. The sensor is loaded by the SANS universal testing machine and the loading process is controlled by the testing machine console. The sensor is connected to the fiber Bragg grating demodulator through the optical fiber and FC/APC connector and then the demodulator is connected to the PC through the network cable.

The wavelength acquisition and identification of the fiber grating adopts the sm125 four-channel demodulator produced by Microoptic Company. This demodulator has a wavelength resolution of 1 pm and a scanning range of 1510 to 1590 nm. The grating used in the sensor is written directly into the single-mode fiber in the core by means of a UV laser. The gate length is 10mm and the initial center wavelengths are 1529.939 (FBG1), 1534.659 (FBG2), 1539.941 (FBG3), 1550.155 (FBG4), 1557.281 (FBG5) and 1562.465 nm (FBG6).

### 4.2. Experimental Procedure and Results

The sensor is coefficient matrix corrected by uniaxial compression. Under uniaxial loading, the wavelength reflected by the FBG on the corresponding axis will shift. At the same time, the other five directions of the reflected wavelength of the FBG will also be affected and shifted. In the process of calibrating the sensor, it is necessary to maintain a constant temperature while applying a load in the Z-axis direction. During the experiment, the axial load is applied continuously and uniformly at a constant speed through the testing machine console. The axial loading process is from 0 to 50 MPa, with a controlled load change rate of 0.125 MPa/s. The variation of the FBG wavelength with stress loading is shown in Figure 12.

It can be seen from the data in Figure 12 and the fitting formula that each FBG reflected wavelength datum fits the stress value with a high degree of linearity. At the same time, the coefficients of determination are all above 0.99, indicating that the sensors have good linearity for uniaxial stress measurements. The sensitivity of the fiber optic gratings 1 to 6 to axial pressure in the *Z*-axis in this experiment was 8.93, 9.09, 24.86, 9.54, 7.95 and 8.69 pm/MPa, respectively. The coefficient matrix between the three-dimensional stress parameters and the wavelength variation of each grating is corrected by the experimental data. The reflection wavelength variation of each fiber grating of the sensor and the three-dimensional stress formula are shown below:(14)$$\left[\begin{array}{l} \Delta \lambda_{1}\\ \Delta \lambda_{2}\\ \Delta \lambda_{3}\\ \Delta \lambda_{4}\\ \Delta \lambda_{5}\\ \Delta \lambda_{6} \end{array}\right]=0.001\times \left[\begin{array}{cccccc} 25.51 & -8.93 & -8.93 & 0 & 0 & 0\\ -9.09 & 25.97 & -9.09 & 0 & 0 & 0\\ -8.7 & -8.7 & 24.86 & 0 & 0 & 0\\ 8.59 & 8.59 & -9.25 & -35.68 & 0 & 0\\ -8.56 & 7.95 & 7.95 & 0 & -33.02 & 0\\ 8.69 & -9.36 & 8.69 & 0 & 0 & -36.10 \end{array}\right]\left[\begin{array}{l} \sigma_{x}\\ \sigma_{y}\\ \sigma_{z}\\ \tau_{xy}\\ \tau_{yz}\\ \tau_{zx} \end{array}\right]$$

According to Equation (14), the sensitivity of the sensor to *X*, *Y* and *Z* axis stress is 25.51, 25.97 and 24.86 pm/MPa, respectively, which is slightly less than the theoretical calculation sensitivity. According to Equation (14) and the measured wavelength variation of the FBG during the calibration experiment, the stress value measured by the sensor is calculated and the comparison with the theoretical value is shown in Table 4.

The error between the calculated stress value and the theoretical value is less than 4%, which can meet the measurement requirements. The reasons for the errors in the experiments are: small size errors occur during the processing of the sensor matrix; theoretical analysis doesn’t consider the influence of slotting on the sensor strain distribution; micro bubbles in the colloid affect the strain transfer during the sensor processing.

## 5. Feasibility of 3D Stress Monitoring of Roadway Surrounding Rock

Under the influence of geological tectonic movement and a self-gravity stress field, the surrounding rock of the deep roadway in a coal mine is in a high ground stress state. The high ground stress leads to the loose and broken rock mass structure of a roadway, the development characteristics of joints and fissures and strong rheology. Due to the strong rheology of the surrounding rock, the borehole constructed in the roadway will be gradually compacted and restored to the original stress of the surrounding rock. At the same time, the surrounding rock stress of the underground roadway increases in a gradient state with the increase of depth. On this basis, accurately measuring the stress distribution law, size and direction of roadway surrounding rock through monitoring equipment, and timely mastering of the stress state information of roadway surrounding rock is an effective means to reduce roadway roof accidents and rock burst disasters.

The three-dimensional stress monitoring system is established according to the principle diagram of an underground coal mine, as shown in Figure 13. The FBG three-dimensional stress sensor is embedded in the borehole using the directional device and fixed by grouting. The three-dimensional stress sensor is connected with the FBG demodulator through the tail fiber, terminal box and communication optical cable. The fiber Bragg grating demodulator is connected with the upper computer and the host computer to monitor and display the reflection center wavelength of the sensor. Then the data is transmitted to the ground server and client computer by the underground Ethernet ring network to realize the remote real-time monitoring and display of the three-dimensional stress state of the roadway surrounding rock. 

According to the rheological stress recovery method, the physical properties of the surrounding rock at the measured point, the direction of the sensor burial and the monitoring data of the sensor, we can calculate the three-dimensional stress state parameters at that point. If we bury several three-dimensional stress sensors in the mining space, combined with mathematical statistics and mathematical analysis methods, we can realize the monitoring of the distribution and evolution of the three-dimensional stress field in the entire mining space, which proves that the FBG three-dimensional stress sensor is feasible to monitor the three-dimensional stress of roadway surrounding rock.

## 6. Conclusions

(1) This paper presents the design principle of a three-dimensional stress sensor for surrounding rocks and derives the numerical relationship between the strain measured by each fiber grating and the three-dimensional stress of the three-dimensional stress sensor. Then, combined with the theory of fiber grating strain measurement, the coefficient matrix between the reflected wavelength variation of each fiber grating and the three-dimensional stress is deduced. Finally, through this matrix, the sensitivity formula of the three-dimensional stress sensor to measure the axial stress of *X*, *Y* and *Z* is obtained.

(2) It is verified by ANSYS Workbench numerical simulation software that, in the uniaxial compression experiment of the sensor, each fiber grating produces cooperative deformation with the matrix strain. At the same time, there is a linear relationship between the deformation of the fiber grating and the uniaxial compression load. By simulating the application of hydrostatic pressure, it can be seen that the groove on the sensor surface has little effect on the strain distribution of the sensor, which verifies the theory of the strain transfer efficiency of the fiber grating. Finally, the three-dimensional stress value obtained by numerical simulation is slightly smaller than the theoretical value and the error is less than 1%, which is consistent with the theoretical analysis.

(3) Through the uniaxial compression experiment to test and calibrate the sensor, we can deduce that the linearity of each fiber grating of the sensor is above 0.99 under the action of uniaxial compression load. We corrected the coefficient matrix with experimental data. From this, it can be calculated that the stress sensitivity of the sensor to the *X*, *Y* and *Z* axes is 25.5, 25.6 and 25.9 pm/Mpa, respectively. Comparing the obtained experimental data with the theoretical value, the relative error is less than 4% which can meet the requirements of the measurement accuracy of the surrounding rock stress.

(4) Due to the limited workload, the influence of temperature on the sensor is not considered in the sensor design and the influence of temperature on the FBG sensor cannot be ignored. The temperature compensation design can be carried out in subsequent research. Secondly, the FBG of this sensor is difficult to effectively protect in a practical application. Subsequent research can consider how to effectively protect the FBG without greatly reducing the accuracy. In addition, in order to enhance the timeliness and visualization, follow-up research can develop supporting three-dimensional stress monitoring data processing software of FBG surrounding rock and directly output the three-dimensional stress value, principal stress value and orientation according to the reflection wavelength of the FBG collected by the computer.

## Figures and Tables

**Figure 1 sensors-22-02624-f001:**
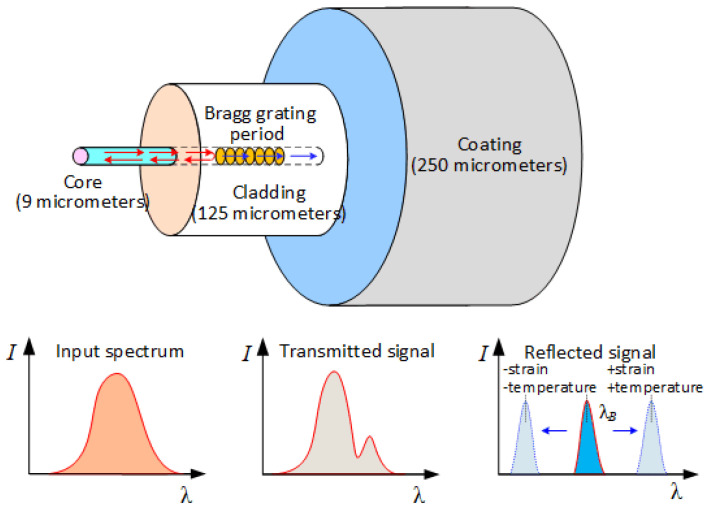
Working principle of fiber Bragg grating.

**Figure 2 sensors-22-02624-f002:**
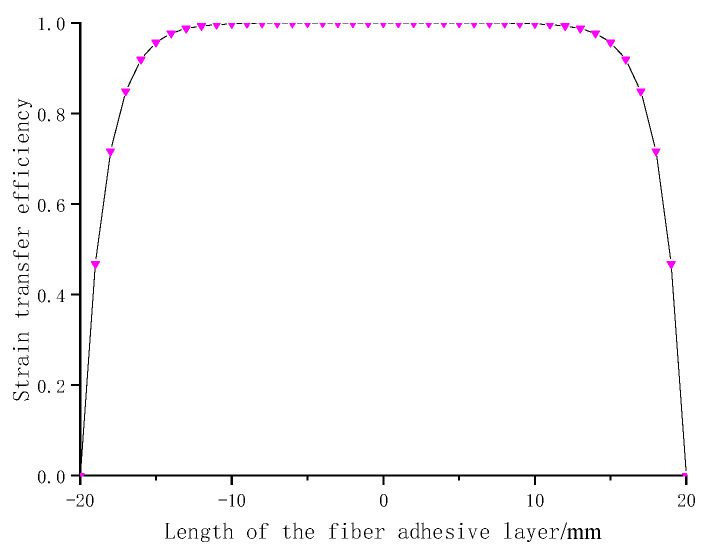
Axial strain transfer efficiency distribution along the length of FBG (pasting length 40 mm).

**Figure 3 sensors-22-02624-f003:**
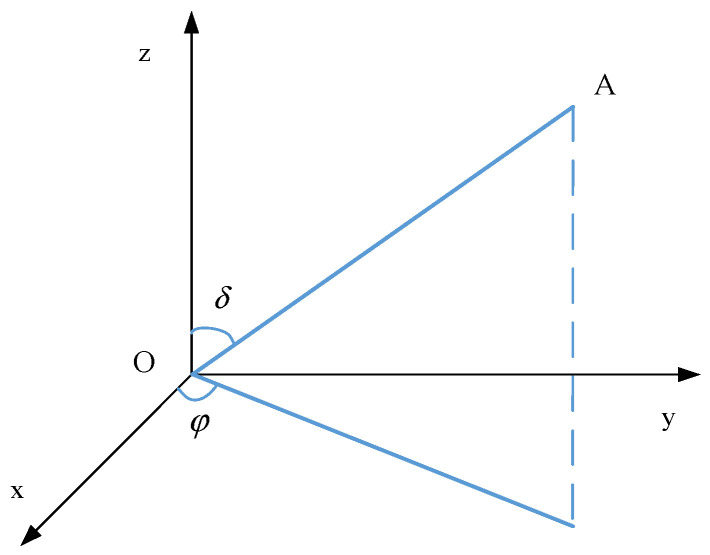
A linear OA in three-dimensional space.

**Figure 4 sensors-22-02624-f004:**
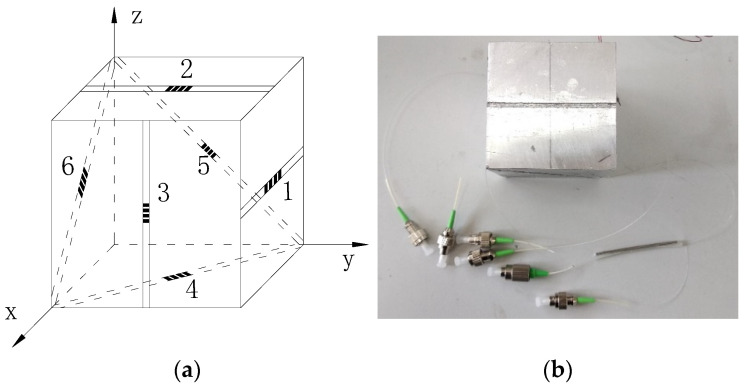
Structure diagram of FBG three-dimensional stress sensor. (**a**) Layout diagram of FBG; (**b**) Physical drawing of the sensor.

**Figure 5 sensors-22-02624-f005:**
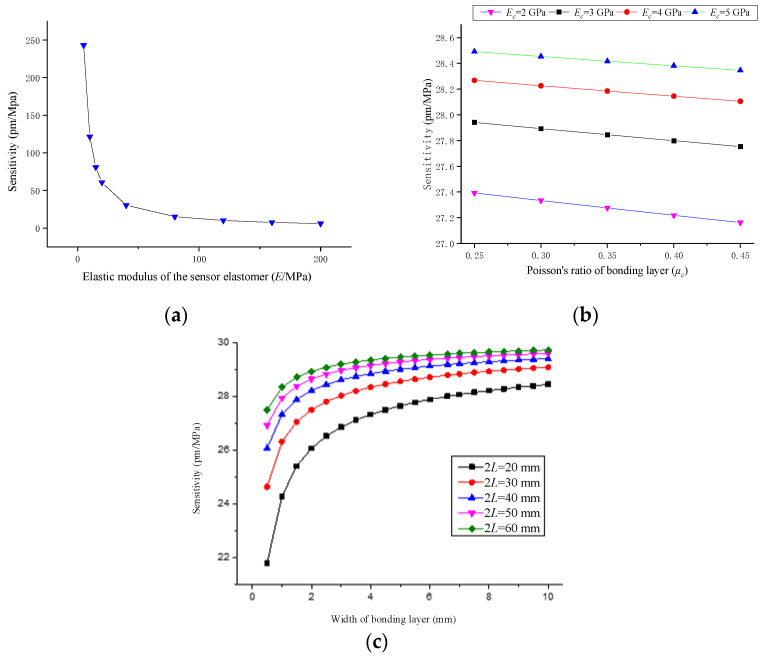
Sensor sensitivity and influence parameter coefficient curve. (**a**) Influence of Matrix Elastic Modulus on Sensitivity: Calculated parameters: 2*L* = 40 mm, *D* = 1 mm, *r_m_* = 1 mm, *E_c_*= 3 GPa, *μ_c_* = 0.3; (**b**) Influence of Elastic Modulus and Poisson’s Ratio of Bonding Layer on Sensitivity: Calculated parameters: 2*L* = 40 mm, *D* = 1 mm, *r_m_* = 1 mm, *E* = 40 Gpa; (**c**) Influence of adhesive layer width and fiber bonding length on sensitivity: Calculated parameters: *r_m_* = 1 mm, *E* = 40 GPa, *E_c_* = 3 GPa, *μ_c_* = 0.3.

**Figure 6 sensors-22-02624-f006:**
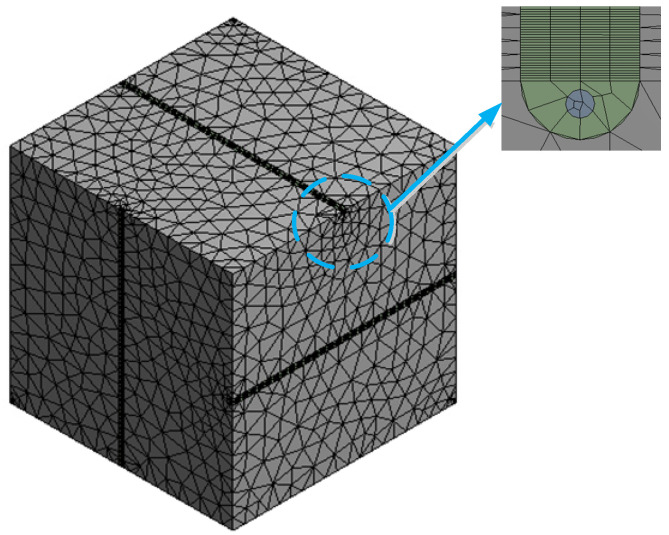
Finite element calculation model.

**Figure 7 sensors-22-02624-f007:**
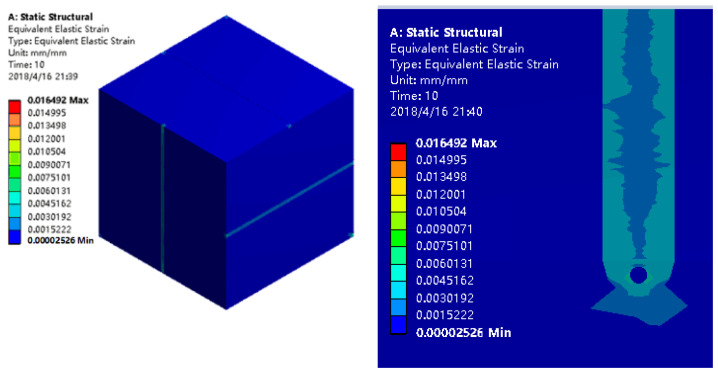
The equivalent strain of the sensor under 50 MPa uniaxial load.

**Figure 8 sensors-22-02624-f008:**
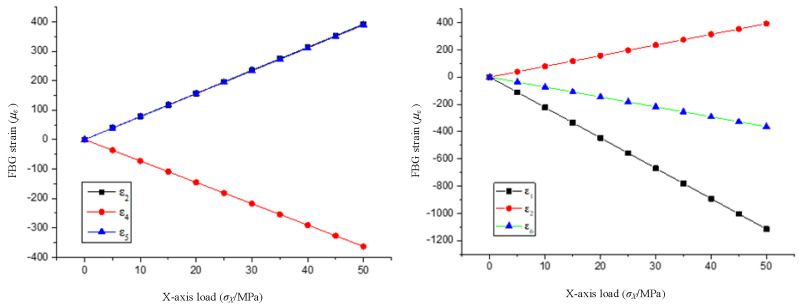
The variation of FBG strain with *X*-axis load.

**Figure 9 sensors-22-02624-f009:**
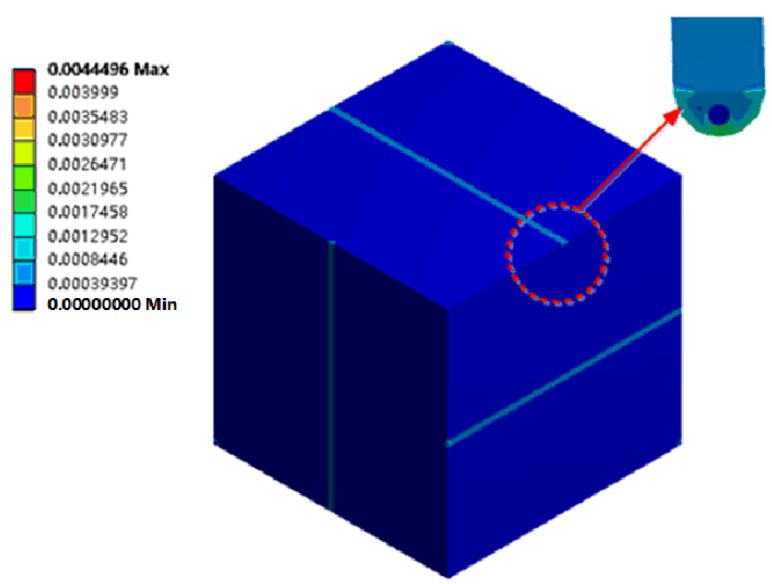
The equivalent strain of the sensor under hydrostatic pressure 50 MPa (The value on the graph is *ε*).

**Figure 10 sensors-22-02624-f010:**
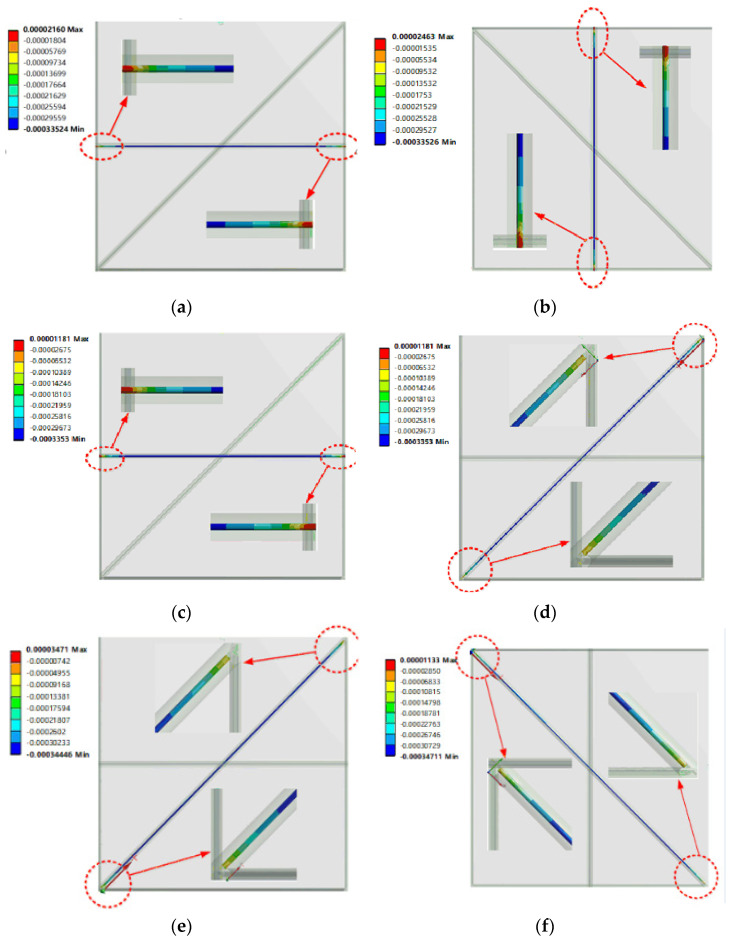
Axial strain distribution of each optical fiber under 50 MPa hydrostatic pressure (The value on the graph is *ε*). (**a**) Strain of optical fiber 1; (**b**) Strain of optical fiber 2; (**c**) Strain of optical fiber 3; (**d**) Strain of optical fiber 4; (**e**) Strain of optical fiber 5; (**f**) Strain of optical fiber 6.

**Figure 11 sensors-22-02624-f011:**
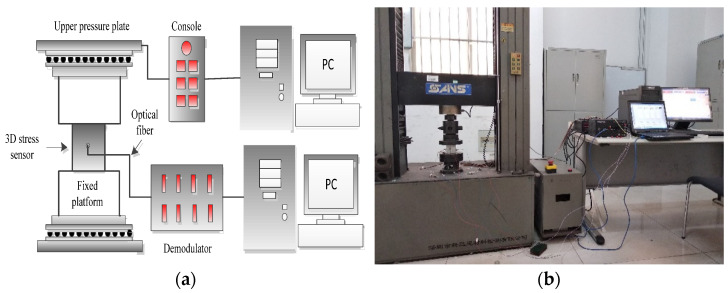
Calibration and test system of the sensor. (**a**) System schematic diagram; (**b**) Physical picture of experiment.

**Figure 12 sensors-22-02624-f012:**
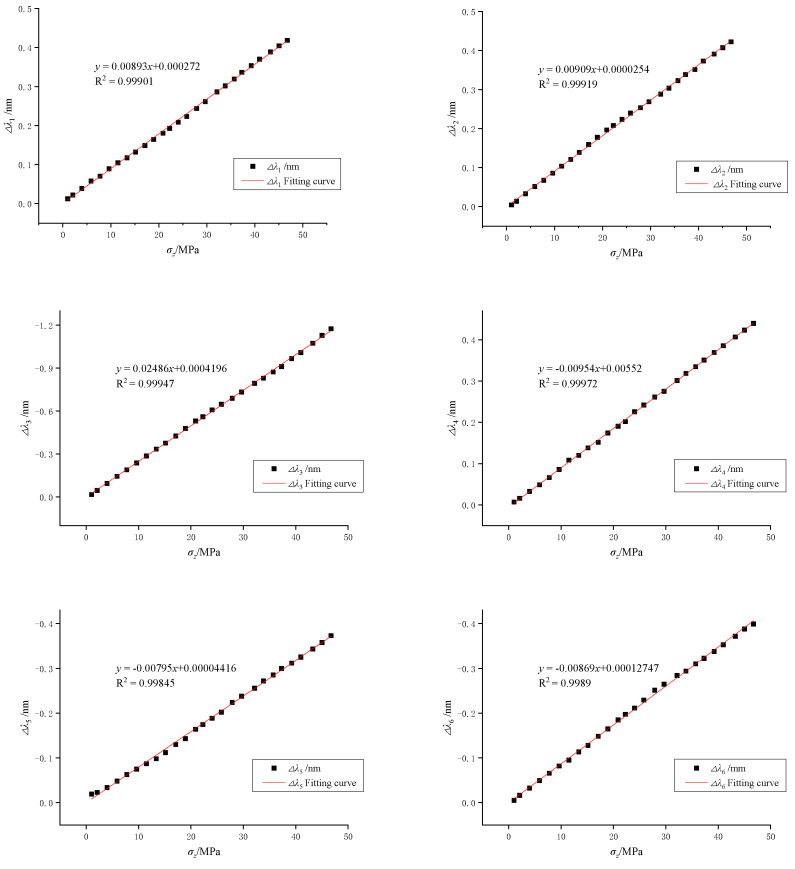
Variation of FBG wavelength with stress.

**Figure 13 sensors-22-02624-f013:**
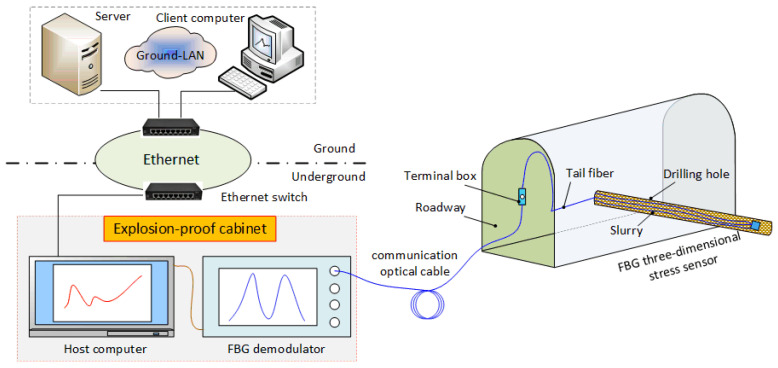
Schematic diagram of surrounding rock stress measurement.

**Table 1 sensors-22-02624-t001:** Structural material parameters of FBG three-dimensional stress sensor.

Symbol	Quantity	Value
*λ* _*B*1_	Center wavelength of the FBG1	1529.939 nm
*λ* _*B*2_	Center wavelength of the FBG2	1534.659 nm
*λ* _*B*3_	Center wavelength of the FBG3	1539.941 nm
*P_e_*	Effective elastic optical coefficient of optical fiber	0.78
*E*	Elastic modulus of the sensor elastomer	44.8 Gpa
2*L*	Length of optical fiber paste	40 mm
*D*	Width of bonding layer	1 mm
*E_c_*	Elastic modulus of bonding layer	3 GPa
*μ_c_*	Poisson’s ratio of bonding layer	0.3
*E_f_*	Elastic modulus of optical fiber	72 Gpa
*r_f_*	Fiber layer radius of sensing structure	0.125 mm
*r_m_*	Elastomer layer radius of sensing structure	1 mm

**Table 2 sensors-22-02624-t002:** The direction cosines of six strain measuring lines in different directions of a three-dimensional stress sensing structure.

Number	*δ*/(°)	*φ*/(°)	*l*	*m*	*n*
1	90	0	1.0	0	0
2	90	90	0	1	0
3	0	90	0	0	1
4	90	135	−0.707	0.707	0
5	135	90	0.0	0.707	−0.707
6	135	0	−0.707	0	−0.707

**Table 3 sensors-22-02624-t003:** Sensor material parameters.

Material	Elastic Modulus/GPa	Poisson Ratio	Density/g/cm^3^
Optical fiber	72	0.25	1.42
Mg-Al alloy	44.8	0.35	1.78
Epoxy resin	4	0.37	1.14

**Table 4 sensors-22-02624-t004:** Comparison between the experimental calculated value and theoretical value.

*Z* Direction Uniaxial Pressure/KN	$\Delta \lambda_{1}/\mathrm{nm}$	$\Delta \lambda_{2}/\mathrm{nm}$	$\Delta \lambda_{3}/\mathrm{nm}$	$\Delta \lambda_{4}/\mathrm{nm}$	$\Delta \lambda_{5}/\mathrm{nm}$	$\Delta \lambda_{6}/\mathrm{nm}$	Calculated Value *σ_z_*/MPa	Theoretical Value *σ_z_*/MPa	Relative Error
9.3667	0.05797	0.05162	−0.1442	0.04879	−0.04836	−0.04936	−5.6561	−5.8538	0.034
21.3641	0.11775	0.12117	−0.33465	0.12026	−0.09838	−0.11382	−13.5799	−13.3526	0.017
30.2294	0.16474	0.17781	−0.47846	0.17434	−0.14341	−0.16519	−19.3861	−18.8934	0.026
38.4442	0.20893	0.22374	−0.60838	0.22559	−0.18877	−0.2117	−24.7468	−24.0276	0.030
51.3686	0.28677	0.28858	−0.79391	0.301421	−0.25618	−0.28451	−31.9275	−32.1054	0.006
62.7429	0.35408	0.35156	−0.96688	0.36932	−0.31217	−0.33757	−38.7162	−39.2143	0.012
72.0102	0.40467	0.4076	−1.1288	0.42376	−0.35785	−0.38813	−45.5894	−45.0064	0.013

## Data Availability

All data and code used or analyzed in this study are available from the corresponding author on reasonable request.

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
