# Peer review of "Research on Three-Dimensional Stress Monitoring Method of Surrounding Rock Based on FBG Sensing Technology"

_sensors, 2022, doi:10.3390/s22072624_

Round 1
Reviewer 1 Report
The authors propose an FBG sensor design, three-dimensional stress sensor for surrounding rocks, to develop a relationship between three-dimensional stress and strain measured by the FBGs. Numerical simulation and experimental compression tests were performed.
This topic has merit and is a well-written manuscript. However, a few comments need to be addressed.
Comment 1:
pg. 3, lines 116-12
Section 2.2 A figure should be added to support equation 4. This will help visually clarify the axial strain of fiber, axial strain of the matrix and length of the fiber adhesive layer relationship of this important concept.
Comment 2:
pg.8, Figure 4
Figure 4a, higher resolution for the x-axis is needed to interpret the sensitivity and Elastic modulus plot. For example, the x-axis is spaced by 50 units.
pg. 15 Conclusions, pg. 365
A couple of statements on future work should be included.
Comment 3:
The grammar in manuscript should be checked.
Author Response
请参阅附件。

Reviewer 2 Report
The manuscript "Research on three-dimensional stress monitoring method of surrounding rock based on FBG sensing technology" shows the proof of concept of the three dimensional stress sensor based on the fiber optic Bragg Gratings.
The work shows both theoretical, including MES simulations, and experimental investigation which shows the principle of operation and practical stress monitoring possibility. The paper is very interesting and well written engineering work, but the novelty of the method is doubtful. I would like to ask authors to describe the novelty of presented solution from the point of view of the measurement potential of the method. What is the advantage of presented sensor in comparison to other methods or FBG based examples (for example https://doi.org/10.3390/app12041781 or https://doi.org/10.1063/5.0077651)
I also have a concern about the assumption of constant temperature in both theoretical and practical models. It is known that the temperature have a huge impact on the FBG operation and in proposed conditions temperature may change. Please comment the method for temperature compensation. Ther are also some errors which should be corrected:
• Figure 11 - on the x-axis the unit is presented in /mm, i think that authors means rather /nm
• Figures 8 and 9 - there are no names of the values or units on the graphs (are they ε, με?)
Round 2
Reviewer 2 Report
The authors addressed all my comments and improve the manuscript in satisfying way. I can recommend the manuscript for publication in Sensors journal.